# Reproductive Biology of the Golden Cuttlefish *Sepia esculenta* (Cephalopoda, Sepiida)

Elizaveta V. Vlasova [1,*] , Rushan M. Sabirov [1] and Alexey V. Golikov [2]

1 Department of Zoology, Kazan Federal University, 420008 Kazan, Russia
2 GEOMAR Helmholtz Centre for Ocean Research Kiel, 24105 Kiel, Germany; golikov.ksu@gmail.com
* Correspondence: evlasova.uni@gmail.com

**Abstract:** The golden cuttlefish *Sepia esculenta* is the one of most abundant cuttlefish species around south-east Asia and has a high commercial value. Despite its wide distribution and high commercial value, its reproductive biology is still poorly understood. This study was based on 25 males and 6 females. The potential fecundity (PF) of females was 1701–3719 oocytes, which was an increase, as compared to the previously known values. The oocyte resorption reached up to 13.2% of fecundity. The ovulation pattern was group-synchronous, with a predominance of previtellogenic oocytes. The pre-meiotic and primary growth oocyte phases were absent in mature females. The number of spermatophores carried by an individual male was 146–1698 (length 9–20 mm). The spermatophores were characterised by a cement body consisting of conical oral and cylindrical aboral parts. The ontogenetic changes in the spermatophores and their parts were recorded for the first time in the order of Sepiida. Their sperm content and their adhesive abilities also increased during ontogenesis. The data obtained in the present study significantly increased and corrected the existing knowledge of *S. esculenta* biology. Moreover, these data help to explain the general patterns of reproductive biology in cuttlefish, as well as in Cephalopoda as a whole.

**Keywords:** commercial invertebrates; reproduction; Indo-Pacific; spermatophores; spermatophorogenesis; spermatangia; oocytes; oogenesis; fecundity; Mollusca





## 1. Introduction

Cuttlefish are cephalopods of small-to-medium size that belong to the order Sepiida and family Sepiidae [1]. Their mantle length (ML) ranges from 20 to 560 mm [2]. Cuttlefish are widely distributed in the temperate and tropical waters of surrounding Europe, Asia, Africa, the Western Pacific, and Australia, where they inhabit the continental shelf and upper slopes, up to a maximum depth of 1000 m [1,3]. The highest concentration of the species was recorded in the Indian Ocean [4]. Many species of cuttlefish have a high commercial value [1]. The global commercial catch of cuttlefish and sepiolid in 2018 reached 348,000 tonnes [5].

Despite the wide distribution of cuttlefish and their high commercial value, data on their reproductive biology are scarce. Most studies connected to the reproductive biology of cuttlefish have been focused on the following: the determination of the sex ratio; spawning seasons; their size at first maturity and sizes at different maturity stages; the monthly distribution of maturity stages; the calculation of certain reproductive indices; and the development of maturity scales for individual species [6–24]. Female reproductive biology has been studied in more detail than that of males. There is a vast amount of data on the fecundity of different species [6,7,9,11,13,16,18,20,23,25–28]; their oogenesis [19,29]; their oocyte phase and size distribution [7,18,25,26,28,30]; their oocyte diameter [11,20,27]; their ovulation patterns [13,27,28]; and oocyte resorption [26]. At the same time, our knowledge of male reproductive biology has been largely limited to data on the number of spermatophores and spermatophore lengths [7,8,16,18,27].

The golden cuttlefish *Sepia esculenta* Hoyle, 1885 [31], is distributed from Central Honshu to the Philippine Islands [1,31,32]. It is the dominant *Sepia* species around the Shantung and Kiangsu Provinces of China and off the coasts of western Japan [1]. The maximum ML is 180 mm [1,33]. Males reach maturity at an ML of 95–115 mm, and females at an ML of 106 mm can contain ripe eggs in the oviduct. The spawning season of *S. esculenta* varies depending on the region: from early March to mid-June in Tokyo Bay; from late March to early July in Mikawa Bay; from early May to early June in the East China Sea; and from June to July in the Yellow Sea [32]. *S. esculenta* has high commercial value and is abundantly caught in the East China Sea [1,33]. Despite its wide distribution and high commercial value, the reproductive biology of *S. esculenta* is poorly understood, and our knowledge currently consists only of the number of spermatophores [34]; the fecundity of the females (Yasuda [35] reported from the coast of Ise and Mikawa Bay and the seas near Atsumi; Tomiyama [36] reported from the Yamaguchi Prefecture coastal waters; Arima et al. [37] reported from the Fukuoka Prefecture coastal waters; and Zhang et al. [38] reported from the Qingdao coastal waters); the ripe egg diameter (reported in the Qingdao coastal waters by Zhang et al. [38]); the oogenesis (data obtained from a cultured population by Yin et al. [29]); and the seminal receptacle's histological structure (reported in the Qingdao coastal waters by Wang et al. [39]). Other studies related to the reproductive biology of the species have focused on reproductive behaviour [40–43]. This study provided data on the morphology of male and female reproductive systems; the number of spermatophores; the spermatophore length and morphology; the ontogenetic changes in spermatophores and spermatophore production; the fecundity of females; the oocyte-phase distribution; the proportion of resorbing oocytes; the histological structure of the ovary; the spermatangia length/structure; and the implantation type of *S. esculenta* in Japanese waters. The new data obtained in this study significantly increase and correct the existing knowledge on the reproductive biology of *S. esculenta*.

## 2. Materials and Methods

### 2.1. Sample Collection, Initial Treatment, Fixation, and Storage

All specimens were captured from shores in Japanese waters: Iriomote Island (24.32° N, 123.93° E; n = 18, all males), Ushimado on Honshu Island (34.62° N, 134.16° E; 3 males and 5 females), and Mukaishima Island (34.37° N, 133.22 °E; 4 males and 1 female). A total of 25 males and 6 females were studied in this work. Detailed information on the studied specimens is presented in Table S1. Cuttlefish were caught with fixed nets or a fishing rod from the shore. Mantle length was measured (up to 1 mm) for 3 males from Ushimado Island, 4 males from Mukaishima Island, and 5 females from Ushimado Island. In females, the ML ranged from 137.0 to 142.0 mm. In males, the ML ranged from 139.0 to 158.0 mm. Body mass was recorded for 3 males from Ushimado Island (256.0–267.0 g). The maturity stages of the reproductive system were determined using the following maturity scale, based on Arkhipkin [44] and Nigmatullin and Sabirov [45]:

Juvenile (0): did not yet have a fully developed reproductive system, was too small to be viewed without a stereomicroscope; reproductive organs were translucent.

Early immature (I): fully formed reproductive system was still small and translucent, but it could be observed by the naked eye.

Late immature (II): reproductive system was larger than at the previous stage; reproductive organs were no longer translucent.

Early maturing (III): reproductive system occupied from one-third to one-half of the volume of the mantle cavity; in males, there was no sperm inside the spermatophoric complex (SC); in females, there were oocytes at several phases.

Late maturing (IV): reproductive system occupied more than a half of the volume of the mantle cavity; in males, sperm presented in the proximal parts of the SC, and the first tentative spermatophores were present; in females, there were vitellogenic oocytes in the ovary.

Pre-mature (V-1): reproductive system was large, similar to the previous stage; in males, up to one-third of the spermatophoric sac was filled by mature spermatophores; in females, first ripe oocytes were present.

Mature (V-2): reproductive system was large, similar to the previous stage; in males, more than one-third of the spermatophoric sac was filled by mature spermatophores; in females, ripe oocytes and possibly postovulatory follicles were present.

Pre-spent (V-3): gonads were undergoing degeneration and of reduced size; in males, the number of spermatophores was the same as at mature stage; in females, residual ripe oocytes were still present.

Spent (VI): in males, testis was undergoing degeneration and of reduced size, and residual spermatophores could be present; in females, the ovary undergoing degeneration and of reduced size, with only resorbing oocytes and postovulatory follicles present.

All studied specimens of both sexes were at V-2 maturity stage.

After the initial treatment, the reproductive systems of all specimens were removed from fresh cuttlefish and fixed in 70° ethanol.

*2.2. Morphological Analysis*

After preservation, the reproductive systems were analysed. In females, the ovary mass was recorded (measured to 0.1 g), and the lengths of the oviduct, the oviducal gland, as well as the nidamental and accessory nidamental glands were measured up to 0.1 mm. The potential fecundity (PF) (i.e., number of oocytes in the ovary and oviduct, if any), oocyte phase distribution, and proportion of oocytes undergoing resorption were counted and assessed by using 5 subsamples from each ovary (mean subsample mass was 0.15 g). Subsamples were weighed, the number of oocytes was counted, and oocyte phases (i.e., previtellogenic, early vitellogenic, mid-vitellogenic, late vitellogenic, and ripe) were determined by visual analysis and measurements. Oocyte diameter (OD) was measured for all oocytes from each subsample. All late vitellogenic and ripe oocytes in the ovary (and ripe in the oviduct, if any) were counted and measured [46,47]. In males, testis and spermatophoric complex (SC) masses were recorded (measured to 0.1 g), and lengths of the testis, sperm duct, spermatophoric glands (SG), and spermatophoric sac were measured to 0.1 mm. A total number of spermatophores were counted in all males, and all spermatophores were measured up to 0.1 mm. A random subsample of 25–40 intact spermatophores was harvested from different sections of the spermatophoric sac and measured in detail. The following measurements were applied: length of head, ejaculatory apparatus, cement body, seminal reservoir and posterior hollow part; maximum width of spermatophore and seminal reservoir; spermatophore mass; and seminal reservoir volume. The volume of the seminal reservoir was calculated using the formula for a volume of a cylinder: $V = 3.14 \times i2 \times l/4$, where $V$ = seminal reservoir volume (mm$^3$), $i$ = seminal reservoir width (mm), and $l$ = seminal reservoir length (mm). The total volume of sperm in all spermatophores carried by an individual male was calculated by multiplying the mean seminal reservoir volume in each section of the spermatophoric sac by the number of spermatophores in the respective section. In females, the seminal receptacles' length and width were measured up to 0.1 mm, and the number of implanted spermatangia was recorded. The spermatangia length and the length of its oral and aboral parts were measured up to 0.1 mm.

The following indexes were calculated for both sexes (where applicable):

Maturity index (MI) = total reproductive system mass × 100/total mass;

Gonadosomatic index (GSI) = gonad mass × 100/total mass;

Ovary index (OvI) = ovary mass × 100/total reproductive system mass;

Spermatophoric complex index − 1 (SCI − 1) = SC mass × 100/total mass;

Spermatophoric complex index $- 2$ (SCI $- 2$) = SC mass $\times$ 100/total reproductive system mass.

Photographic images were taken using an optical microscope, Axio Imager A2, and a digital video microscope, Hirox KH-7700. Scanning electron microscopy examinations were performed using the Hitachi TM-1000. To prepare for the analysis, the spermatophores were washed carefully with distilled water, dehydrated using ascending ethanol concentrations (70%, 80%, 90%, 96%, and 100%), and then $CO_2$ critical-point dried.

*2.3. Histological Analysis*

The testes of three males and the ovaries of three females were examined using histological methods. To obtain histological sections, the material was embedded in paraffin blocks. The material was preliminarily dehydrated by passing through a graded series of ethanol solutions. Before embedding, the material was dehydrated using ascending ethanol concentrations (70%, 80%, 90%, 96%, and 100%), then by a mixture of 100% ethanol with xylene, xylene, and xylene–paraffin, and paraffinated in pure paraffin [48]. The sections were obtained on a rotary microtome Microm HM 325. Staining was carried out with haematoxylin and eosin.

*2.4. Statistical Analysis*

The differences among the studied groups were assessed by dispersion analysis [49]. The Mann–Whitney U-test was used when comparing the two groups, and the Kruskal–Wallis H test was used when comparing more than two groups. A regression analysis was used to identify equations fitting our data [49]. A value of $\alpha = 0.05$ was taken as significant in this study. The following software were used for statistical data processing: PAST v. 3.15 [50] and MS Excel 2010. The values are given as minimum to maximum (mean $\pm$ SE), unless otherwise stated.

**3. Results**

*3.1. Female Reproductive System*

3.1.1. Morphology of the Reproductive System

The female reproductive system of *S. esculenta* consisted of a single ovary, a left oviduct and oviducal gland, and paired nidamental and accessory nidamental glands (Figure 1a,b). The total mass of the reproductive system ranged from 18.4 to 21.5 (20.1 $\pm$ 0.44) g, and the MI was 7.1–8.3 (7.4 $\pm$ 0.23)%.

The ovary was located in the visceral–pericardial coelom in the posterior part of the mantle cavity. The ovary mass ranged from 6.4 to 8.4 (7.1 $\pm$ 0.36) g; the GSI was 2.4–3.01 (2.6 $\pm$ 0.12)%, and the OvI was 29.6–40.6 (35.3 $\pm$ 1.92)%. The oviduct (Figure 1b) was located in the left part of the mantle cavity, under the left nidamental gland, and looked like a transparent tube. The 4 mature females had 2–8 ripe oocytes in the oviduct. The oviducal gland (Figure 1b) was located at the distal end of the oviduct. It consisted of distal and proximal parts. The proximal part was oval and homogeneous; the distal part was elongated and consisted of many lamellae. The proximal part was more massive than the distal one. The nidamental glands (Figure 1a) occupied the entire volume of the central part of the mantle cavity. They were pear shaped and consisted of many thin lamellae with a central excretory duct. The right nidamental gland was slightly larger than the left one since the oviduct was located under the left gland, compressing it. The accessory nidamental glands (Figure 1a) were of almost rectangular form. The right accessory nidamental gland was slightly larger than the left one. The detailed measurements of all parts of the female reproductive system are provided in Table S2.

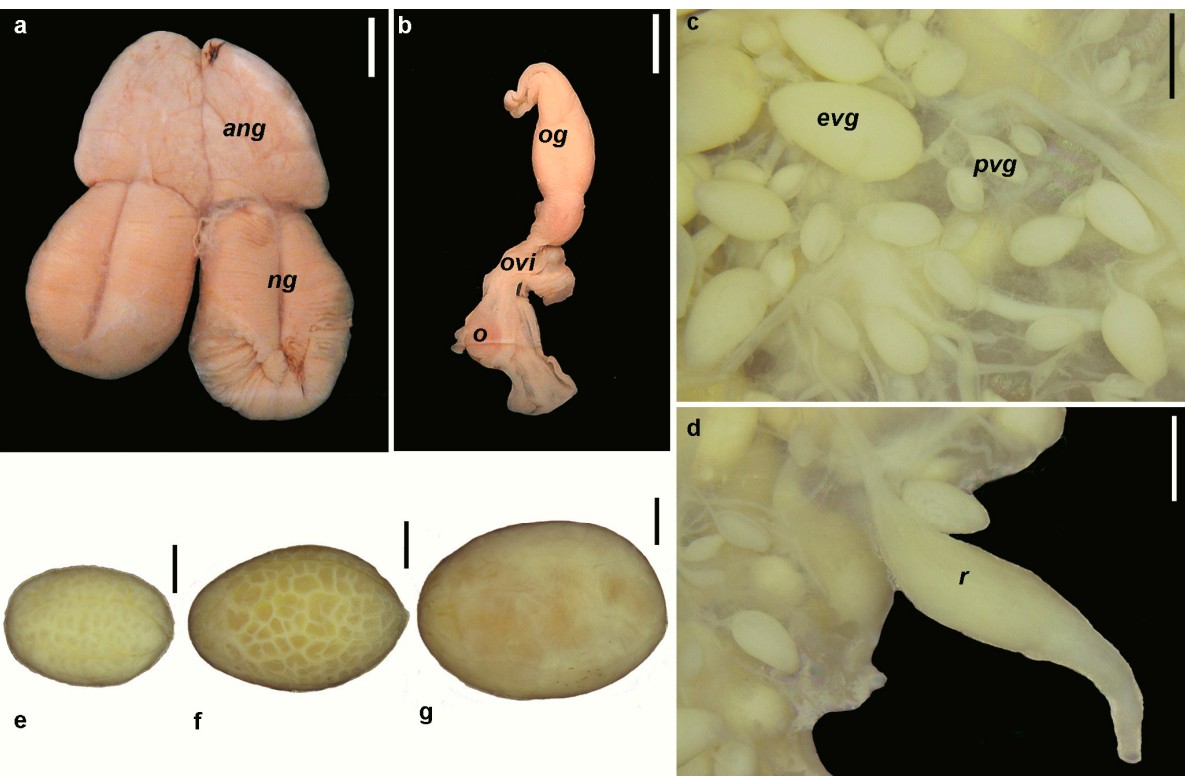

**Figure 1.** External view of reproductive system and oocytes of *Sepia esculenta*: (**a**) Nidamental glands and accessory nidamental glands; (**b**) Oviduct with ripe oocyte; (**c**) Oocytes in the ovary; (**d**) resorbed oocyte in the ovary; (**e**) mid-vitellogenic oocyte; (**f**) late vitellogenic oocyte; and (**g**) ripe oocyte. *ang*: accessory nidamental glands; *ng*: nidamental glands; *og*: oviducal gland; *ovi*: oviduct; o: ripe oocyte; *evg*: early vitellogenic oocyte; *pvg*: previtellogenic oocyte; *r*: resorbed oocyte. Scale bars: (**a–b**): 10 mm, (**c–g**): 1 mm.

### 3.1.2. Potential Fecundity

The PF of females ranged from 1701 to 3719 oocytes (2945.1 ± 317.30). There was a quantity ratio of oocyte phases present in the ovary, as most of the oocytes were in the previtellogenic phase (54.6–88.1%), and the rest were distributed as follows: 6.6–11.3% in the early vitellogenic phase, 1.1–8.2% in the mid-vitellogenic phase, and 1.5–6.8% in the late vitellogenic phase (Figure 2). The ripe oocytes in the ovary numbered from 1 to 14, and 0–8 ripe oocytes were observed in the oviduct. The resorbing oocytes ranged from 0.6 to 13.2 (8.7 ± 1.76)% PF. The diameters of the oocytes undergoing resorption were 0.2–5.7 (2.9 ± 0.03) mm (Figure 3). The measurements of the different oocyte phases are provided in Table 1.

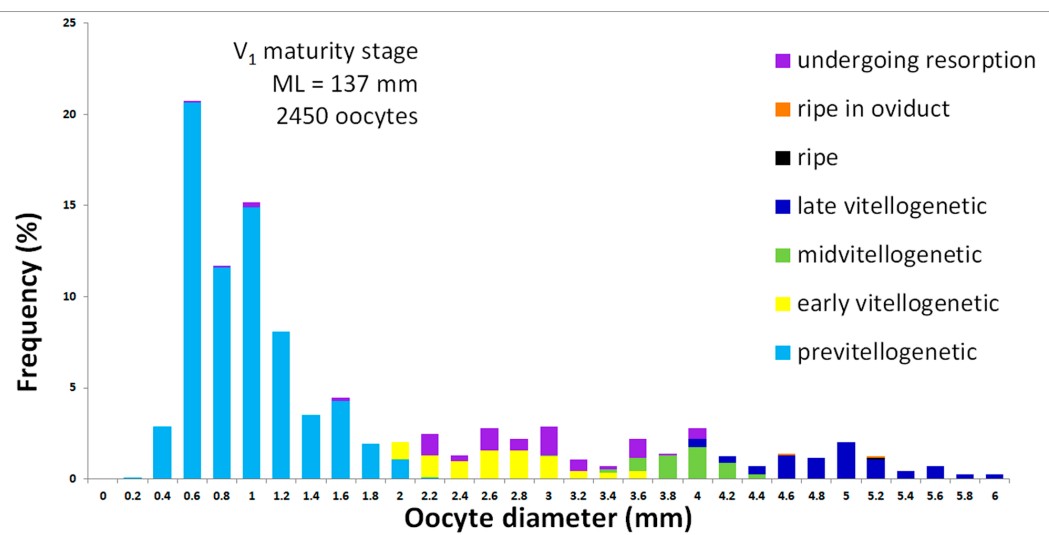

**Figure 2.** Oocyte phases distribution in the ovary of *Sepia esculenta* female.

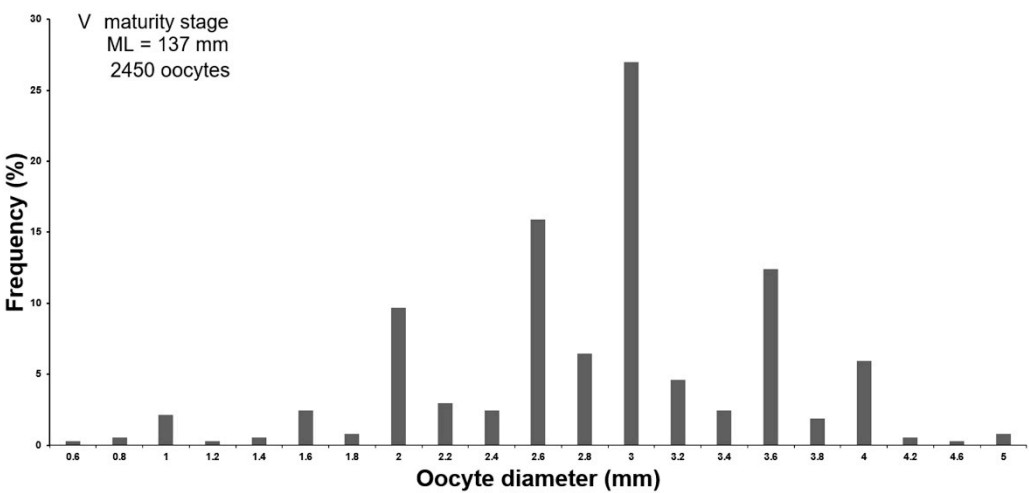

**Figure 3.** Distribution of resorbed oocytes diameters in the ovary of *Sepia esculenta* female.

**Table 1.** Measurements of different oocyte phases in *Sepia esculenta*. ML, mantle length.

| Oocyte Diameter | Previtellogenesis | Early Vitellogenesis | Mid-Vitellogenesis | Late Vitellogenesis | Ripe | Ripe in Oviduct | Resorption |
|---|---|---|---|---|---|---|---|
| Absolute | 0.1–2.5 mm (0.9 ± 0.01 mm) | 0.3–3.9 (2.5 ± 0.02) | 2.3–5.1 mm (3.8 ± 0.02 mm) | 3.6–7.0 mm (4.8 ± 0.02 mm) | 3.6–5.8 mm (5.3 ± 0.08 mm) | 3.0–6.0 mm (4.9 ± 0.21 mm) | 0.2–5.7 mm (2.9 ± 0.03 mm) |
| In % of ML | 0.1–1.8% (0.6 ± 0.01%) | 0.2–2.8% (1.8 ± 0.01%) | 1.6–3.7% (2.7 ± 0.02%) | 2.6–4.2% (3.5 ± 0.02%) | 2.6–4.2% (3.8 ± 0.07%) | 2.1–4.2% (3.5 ± 0.15%) | 0.1–4.1% (2.1 ± 0.02%) |

### 3.1.3. Oogenesis

Seven oocyte phases were observed in the ovaries of mature *S. esculenta* females by histological analyses (Figure 4):

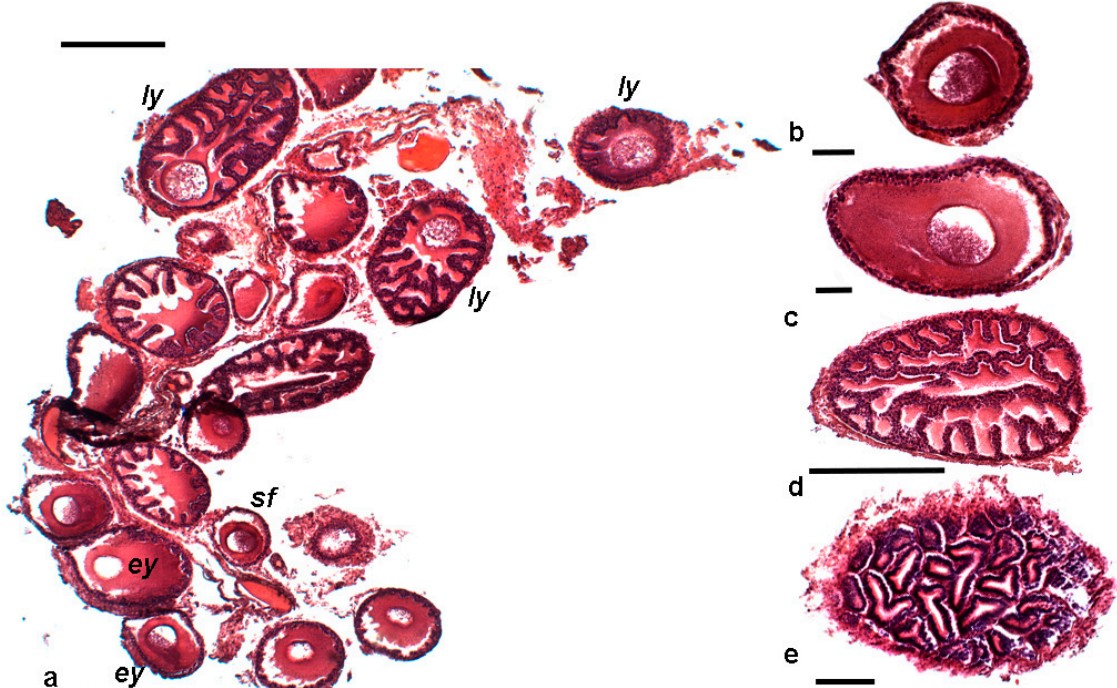

**Figure 4.** Histological section of the ovary of mature *Sepia esculenta* female: (**a**) Oocytes cluster; (**b**) Simple follicle phase; (**c**) Early yolkless phase; (**d**) Late yolkless phase; (**e**) Early vitellogenic phase. *sf*: simple follicle; *ey*: early yolkless; *ly*: late yolkless. Scale bars: (**a**,**d**,**e**): 0.5 mm; (**b**,**c**): 200 μm.

Period of Oogonia Production

Phase 1. Pre-meiotic oocytes

    This stage was absent.

Protoplasmic Growth Period

Phase 2. Primary growth

    This stage was absent.

Phase 3. Simple follicle

    The oocyte shape was rounded or elongated. The nucleus was located in the centre of the cell or slightly shifted to the animal pole. Most of the cell volume was occupied by cytoplasm. The oocyte was surrounded by two layers of follicular cells (Figure 4b).

Interstitial Period

Phase 4. Early yolkless

    The oocyte shape was elongated, and the nucleus was strictly located on the animal pole. The follicular epithelium folds had started to penetrate into the oocyte but did not yet reach the medium line of the oocyte radius. Most of the cell volume was still occupied by the cytoplasm (Figure 4c).

Phase 5. Late yolkless

    The oocyte shape was elongated, and the nucleus was located on the animal pole. Almost the entire cell volume was occupied by the follicular epithelium folds (Figure 4d).

Trophoplasmic Growth Period

Phase 6. Early vitellogenesis

The oocyte shape was elongated, and the nucleus was not observed. Most of the cell volume was occupied by the follicular epithelium folds, and the yolk accumulation had begun. The oocyte sizes ranged from 0.3 to 3.9 (2.5 ± 0.02) mm (Figures 1c and 4e).

Phase 7. Middle vitellogenesis

The oocyte shape was elongated, and the nucleus was not observed. The follicular epithelium folds had reached the medium line of the oocyte radius, and almost the entire volume of the oocyte was filled with yolk. The oocyte sizes ranged from 2.3 to 5.1 (3.8 ± 0.02) mm (Figure 1e).

Phase 8. Late vitellogenesis

The oocyte shape was elongated, and the nucleus was not observed. The follicular epithelium folds occupied less than half of the oocyte volume. The oocyte sizes ranged from 3.6 to 7.0 (4.8 ± 0.02) mm (Figure 1f).

Phase 9. Ripe oocyte

The entire volume of the oocyte was filled with yolk, and the follicular epithelium folds no longer penetrated the cell. The oocyte sizes ranged from 3.0 to 6.3 (5.1 ± 0.10) mm (Figure 1g).

*3.2. Male Reproductive System*

3.2.1. Morphology of the Reproductive System

The male reproductive system of *S. esculenta* consisted of a single testis and the SC. The total mass of the reproductive system ranged from 3.7 to 16.9 (8.5 ± 1.07) g, and the MI was 2.4–3.2 (2.9 ± 0.22)%. The testis was situated in the visceral–pericardial coelom in the posterior part of the mantle cavity. It had a triangular or oval shape and consisted of many seminal tubes tightly adjacent to each other. The testis weight was 1.6–9.8 (4.1 ± 0.57) g, and the GI ranged from 1.2 to 1.6 (1.4 ± 0.12)%. The single SC (Figure 5) was located in the left half of the mantle cavity; it was composed of the sperm duct, six SGs and the spermatophoric sac. The mass of the SC varied from 1.8 to 8.7 (3.9 ± 0.30) g; the SCI–1 was 1.2–1.6 (1.4 ± 0.10)%, and SCI–2 was 53.7–145.0 (102.8 ± 4.82)%.

The first part of the SC was a sperm duct. It consisted of numerous compactly packed coils, along which the sperm could move. The sperm duct opened into the SG. The SG consisted of six morphologically different parts. Parts I and II were tightly adjoined with each other and formed a sac-like structure. Part III had the shape of a slightly curved, short cylinder. Part IV was the longest part of the SG and had a cylindrical shape. Part IV was connected to part V via an intermediate duct. An excretory tube deviated from the intermediate duct and opened freely into the cavity of the genital bag. Part V had the shape of an elongated bag, and Part VI had a loop-like shape. Part VI was connected to a spermatophoric sac via a spermatophoric duct. The measurements of all the parts of the male reproductive system are presented in Table S2.

The spermatophoric sac was divided into three inner portions: the fundus (i.e., the proximal part of the spermatophoric sac), the central part, and the penis (i.e., the distal part of the spermatophoric sac). The fundus contained 1.6–37.1 (16.7 ± 1.94)% of all the spermatophores carried by an individual male. The central part contained 45.6–90.2 (72.2 ± 3.02)% of all the spermatophores carried by an individual male. The penis contained 0.0–45.6 (2.1 ± 2.44)% of all the spermatophores carried by an individual male; 3 males had no spermatophores in the penis. The weight of all the spermatophores contained in the spermatophoric sac was 1.3–7.2 (2.5 ± 0.24) g.

The testis and all six SGs showed negative allometric growth, i.e., the length of the respective parts decreased with increasing ML (Figure S1, Table S3). The spermatophoric sac length showed positive allometric growth, i.e., its length increased with increasing ML (Figure S1, Table S3).

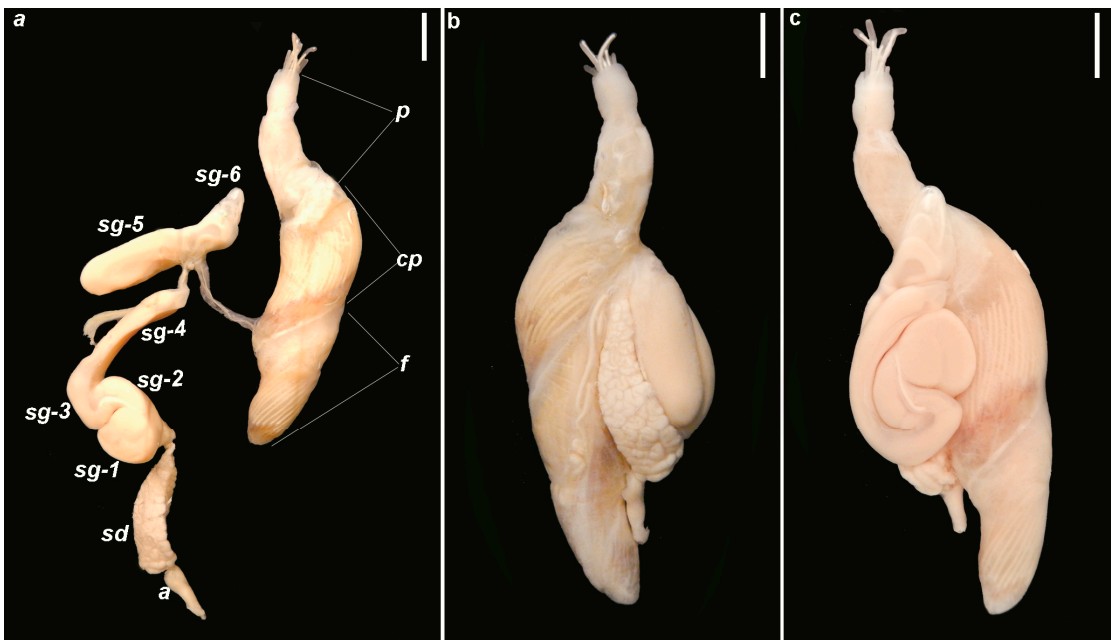

**Figure 5.** External view of spermatophoric complex of *Sepia esculenta*: (**a**) Morphology of spermatophoric complex; (**b**) Dorsal view of spermatophoric complex; (**c**) Ventral view of spermatophoric complex. *a*: ampula; *sd*: sperm duct; *sg-1*: part I of spermatophoric gland; *sg-2*: part II of spermatophoric gland; *sg-3*: part III of spermatophoric gland; *sg-4*: part VI of spermatophoric gland; *sg-5*: part V of spermatophoric gland; *sg-6*: part VI of spermatophoric gland; *f*: fundus; *cp*: central portion of spermatophoric sac; *p*: penis. Scale bars: 10 mm.

### 3.2.2. Number of Spermatophores

The number of the spermatophores carried by an individual male ranged from 146 to 1698 ($422.7 \pm 79.92$). A significant correlation was found between the ML and the number of spermatophores: the number of spermatophores = $7.03 \times \text{ML} - 627.46$; $r^2 = 0.81$; $p = 0.0145$; n = 7 (Figure S2). Additionally, a significant correlation was observed between the number of the spermatophores and the total mass of the reproductive system: the number of spermatophores = $89.18 \times$ total mass of reproductive system $- 238.38$; $r^2 = 0.75$; $p < 0.0001$; n = 17 (Figure S3). The estimated total volume of the sperm in all the spermatophores carried by an individual male varied from 208.0 to 1562.6 ($620.4 \pm 68.66$) mm$^3$. The positive correlation between the number of spermatophores and the total volume of sperm was significant: the total volume of sperm = $0.79 \times$ the number of spermatophores $+ 271.72$; $r^2 = 0.85$; $p < 0.0001$; n = 22 (Figure S4).

### 3.2.3. Spermatophores

The spermatophores were elongated, slightly curved, cylindrical tubes (Figure 6). The spermatophore length (SL) varied among the specimens, from 9.0 to 20.0 ($15.0 \pm 0.02$) mm. The relative spermatophore length ranged from 7.6 to 13.6 ($9.9 \pm 0.02$)% ML. The oral end of the spermatophore had a short head covered by a cup that bore a long spermatophoric thread. The ejaculatory apparatus, a complex system of membrane structures, was approximately equal in length to the head. The cement body was clearly divided into oral and aboral parts. The oral part was conical and approximately 1.5 times shorter than the aboral one. The aboral part was cylindrical. The single seminal reservoir had the shape of a slightly curved cylinder. It was the longest part of the spermatophore and reached up to 32.1–90.0 ($70.4 \pm 0.21$)% SL. The volume of the seminal reservoir ranged from 0.07 to 8.3 ($1.6 \pm 0.03$) mm$^3$. There was a significant correlation between the seminal reservoir volume and the SL: the seminal reservoir volume = $0.28 \times \text{SL} - 2.56$; $r^2 = 0.17$; $p < 0.0001$; n = 752, indicating that the longer spermatophores contained a larger volume of sperm. A well-developed posterior hollow part was present in the aboral region of the

spermatophore. The measurements of all the parts of the spermatophores are presented in Table 2.

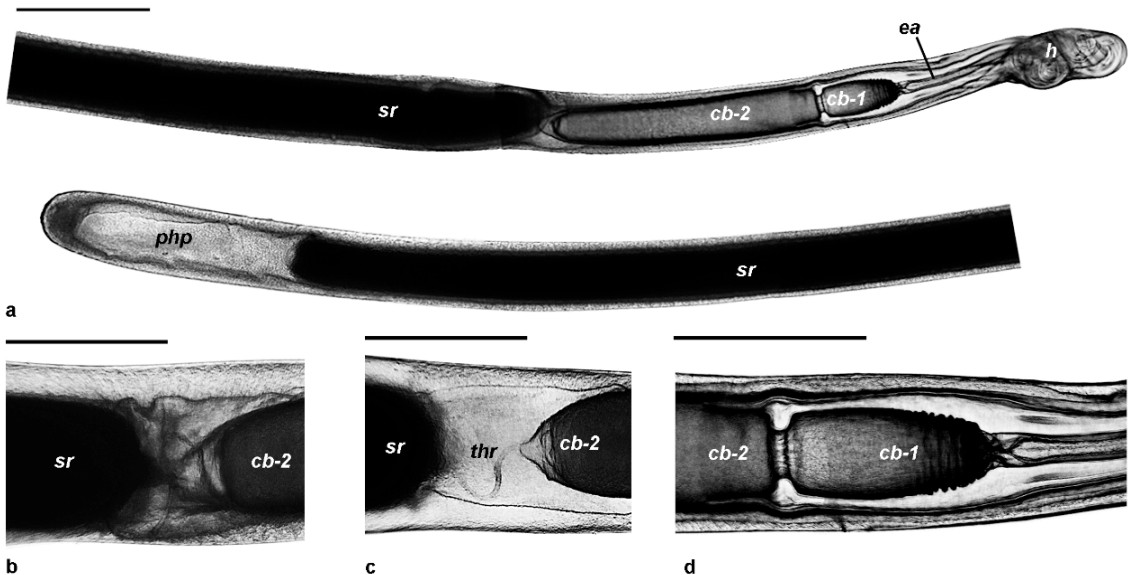

**Figure 6.** Spermatophore morphology of *Sepia esculenta*: (**a**) Entire spermatophore, (**b**,**c**) Connection between cement body and seminal reservoir, (**d**) Connection between oral and aboral parts of cement body. *h*: head; *ea*: ejaculatory apparatus; *cb-1*: oral part of cement body: *cb-2*: aboral part of cement body; *sr*: seminal reservoir; *php*: posterior hollow part; *thr*: thread, that connected seminal reservoir and cement body. Scale bars: (**a**): 1 mm, (**b**,**c**): 0.5 mm.

The following correlations were found among the SL and the measurements of the spermatophore parts: (1) the SL and the relative SL decreased with an increasing ML, and the SL showed a slight tendency to have a negative correlation with the ML when the relative SL had a well-expressed negative correlation with the ML (Figure S5); (2) the head, the cement body, the seminal reservoir, and the posterior hollow part increased in length with an increasing SL; (3) the ejaculatory apparatus length decreased with an increasing SL; and (4) the spermatophore and the seminal reservoir widths increased with an increasing SL (Figure S6, Table S4).

The spermatophores from the different portions of the spermatophoric sac had different mean lengths. The ontogenetic increase in the spermatophore size (i.e., when the spermatophores produced later in ontogenesis were larger than those produced earlier) was clearly observed inside the spermatophoric sac in males that had been collected at the beginning of the spawning season. The mean length of the younger spermatophores from the fundus was 15.29 ± 0.03 mm. The mean length of the older spermatophores from the central portion of the spermatophoric sac was 14.68 ± 0.02 mm, and the mean length of the oldest spermatophores from the penis was 14.39 ± 0.06 mm. The difference in length and width between the younger and older spermatophores was significant (Table 2). Additionally, there were significant differences between the younger and older spermatophores in the spermatophore mass, the seminal reservoir width, the seminal reservoir volume, and the cement body length (Table 2). There were no significant differences between the younger and older spermatophores in head length and ejaculatory apparatus length (Table 2). The difference in the seminal reservoir length between younger and older spermatophores was not significant, but there was a tendency for a decrease in the seminal reservoir length from the fundus to the penis (Table 2). In males collected at the end of the spawning season, the spermatophores from the fundus had a mean length of 14.55 ± 0.11 mm; the spermatophores from the central portion had a mean length of 15.35 ± 0.03 mm; and the spermatophores from the penis had a mean length of 14.34 ± 0.11 mm. The spermatophores from the fundus were smaller than the spermatophores from the central part (Table 2). All three samples

of the spermatophores (from the fundus, central part and penis) differed significantly in length (Table 2). The mean length, width, and volume of the seminal reservoir and the spermatophore weight were also smaller in the fundus than in the central part (Table 2). In contrast, the length of the head, the ejaculatory apparatus, and the cement body showed a tendency to decrease from fundus to penis (Table 2).

**Table 2.** Measurements of spermatophores and spermatophore parts in different portions of spermatophoric sac in *Sepia esculenta*. Significant *p*-values are in **bold**.

| Measurements | Specimens Collected at the Beginning of Spawning Season | | | Specimens Collected at the End of Spawning Season | | |
|---|---|---|---|---|---|---|
| | **Fundus** | **Central Portion** | **Penis** | **Fundus** | **Central Portion** | **Penis** |
| Spermatophore length | 13.0–18.0 mm (15.3 ± 0.03 mm) | 12.0–17.5 mm (14.7 ± 0.02 mm) | 12.0–20.0 mm (14.4 ± 0.06 mm) | 11.15–19.50 mm (14.6 ± 0.11 mm) | 11.0–20.0 mm (15.4 ± 0.03 mm) | 9.0–19.0 mm (14.3 ± 0.11 mm) |
| | $H = 291.6$, $p < 0.0001$ | | | $H = 291.6$, $p < 0.0001$ | | |
| Width of spermatophore | 0.5–0.7 mm (0.54 ± 0.01 mm) | 0.4–0.6 mm (0.53 ± 0.004 mm) | 0.4–0.6 mm (0.51 ± 0.01 mm) | 0.3–0.7 mm (0.45 ± 0.01 mm) | 0.3–0.9 mm (0.48 ± 0.01 mm) | 0.3–0.5 mm (0.40 ± 0.01 mm) |
| | $H = 7.49$, $p = 0.0116$ | | | $H = 32.77$, $p < 0.0001$ | | |
| Weight of spermatophore | 2.2–6.0 mg (3.83 ± 0.07 mg) | 2.5–4.7 mg (3.71 ± 0.04 mg) | 2.4–4.5 mg (3.51 ± 0.07 mg) | 1.1–6.4 mg (2.2 ± 0.13 mg) | 0.9–8.0 mg (2.83 ± 0.09 mg) | 0.9–4.7 mg (1.97 ± 0.15 mg) |
| | $H = 11.84$, $p = 0.0022$ | | | $H = 33.43$, $p < 0.0001$ | | |
| Length of head | 0.6–0.9 mm (0.78 ± 0.01 mm) | 0.6–1.0 mm (0.77 ± 0.01 mm) | 0.4–0.9 mm (0.78 ± 0.01 mm) | 0.3–1.1 mm (0.76 ± 0.03 mm) | 0.3–1.3 mm (0.76 ± 0.02 mm) | 0.4–1.8 mm (0.65 ± 0.04 mm) |
| | $H = 1.23$, $p = 0.49$ | | | $H = 13.19$, $p = 0.0012$ | | |
| Length of ejaculatory apparatus | 0.4–0.9 mm (0.63 ± 0.01 mm) | 0.4–1.1 mm (0.63 ± 0.01 mm) | 0.3–0.9 mm (0.65 ± 0.02 mm) | 0.2–2.3 mm (1.14 ± 0.09 mm) | 0.3–2.3 mm (0.95 ± 0.04 mm) | 0.2–2.9 mm (0.88 ± 0.09 mm) |
| | $H = 2.37$, $p = 0.27$ | | | $H = 3.98$, $p = 0.13$ | | |
| Length of cement body | 2.0–2.7 mm (2.3 ± 0.02 mm) | 1.8–2.7 mm (2.3 ± 0.02 mm) | 1.6–2.5 mm (2.1 ± 0.03 mm) | 0.9–3.9 mm (2.01 ± 0.09 mm) | 0.7–2.7 mm (1.94 ± 0.04 mm) | 0.8–3.0 mm (1.84 ± 0.09 mm) |
| | $H = 21.57$, $p < 0.0001$ | | | $H = 2.99$, $p = 0.22$ | | |
| Length of seminal reservoir | 9.5–11.8 mm (10.56 ± 0.07 mm) | 9.0–11.5 mm (10.38 ± 0.05 mm) | 8.9–11.0 mm (10.31 ± 0.10 mm) | 6.1–12.5 mm (9.16 ± 0.24 mm) | 4.3–14.0 mm (10.41 ± 0.13 mm) | 6.1–13.7 mm (9.67 ± 0.30 mm) |
| | $H = 4.40$, $p = 0.11$ | | | $H = 19.01$, $p < 0.0001$ | | |
| Width of seminal reservoir | 0.3–0.6 mm (0.44 ± 0.01 mm) | 0.3–0.6 mm (0.41 ± 0.01 mm) | 0.3–0.5 mm (0.42 ± 0.01 mm) | 0.1–0.6 mm (0.38 ± 0.02 mm) | 0.2–0.9 mm (0.45 ± 0.01 mm) | 0.2–0.5 mm (0.36 ± 0.01 mm) |
| | $H = 7.53$, $p = 0.0102$ | | | $H = 27.79$, $p < 0.0001$ | | |
| Length of posterior hollow part | 0.1–1.6 mm (0.68 ± 0.06 mm) | 0.1–2.0 mm (0.91 ± 0.04 mm) | 0.2–3.6 mm (0.84 ± 0.09 mm) | 0.1–3.5 mm (0.46 ± 0.08 mm) | 0.1–5.2 mm (0.61 ± 0.06 mm) | 0.1–3.4 mm (0.72 ± 0.13 mm) |
| | $H = 11.44$, $p = 0.0032$ | | | $H = 1.37$, $p = 0.47$ | | |

### 3.2.4. Spermatangia

A total of 6 mature females carried spermatangia ranging from 19 to 37 (27 ± 3.21). The spermatangia were implanted on the seminal receptacle under the buccal membrane (Figure 7a,b). The seminal receptacle length was 15.0–28.0 mm (19.9 ± 1.19) mm, and its maximal width was 8.0–18.0 (12.6 ± 0.82) mm. In addition, 1 female carried 48 spermatangia on the buccal membrane (Figure 7c,f). The length of the spermatangia ranged from 0.7 to 4.4 (3.0 ± 0.12) mm. The spermatangia were divided into oral (0.5–2.0, 1.1 ± 0.06 mm) and aboral (0.7–4.4, 3.0 ± 0.12 mm) parts (Figure 7d). The implantation of spermatangia was shallow, where the oral part had only penetrated into female tissues (Figure 7e).

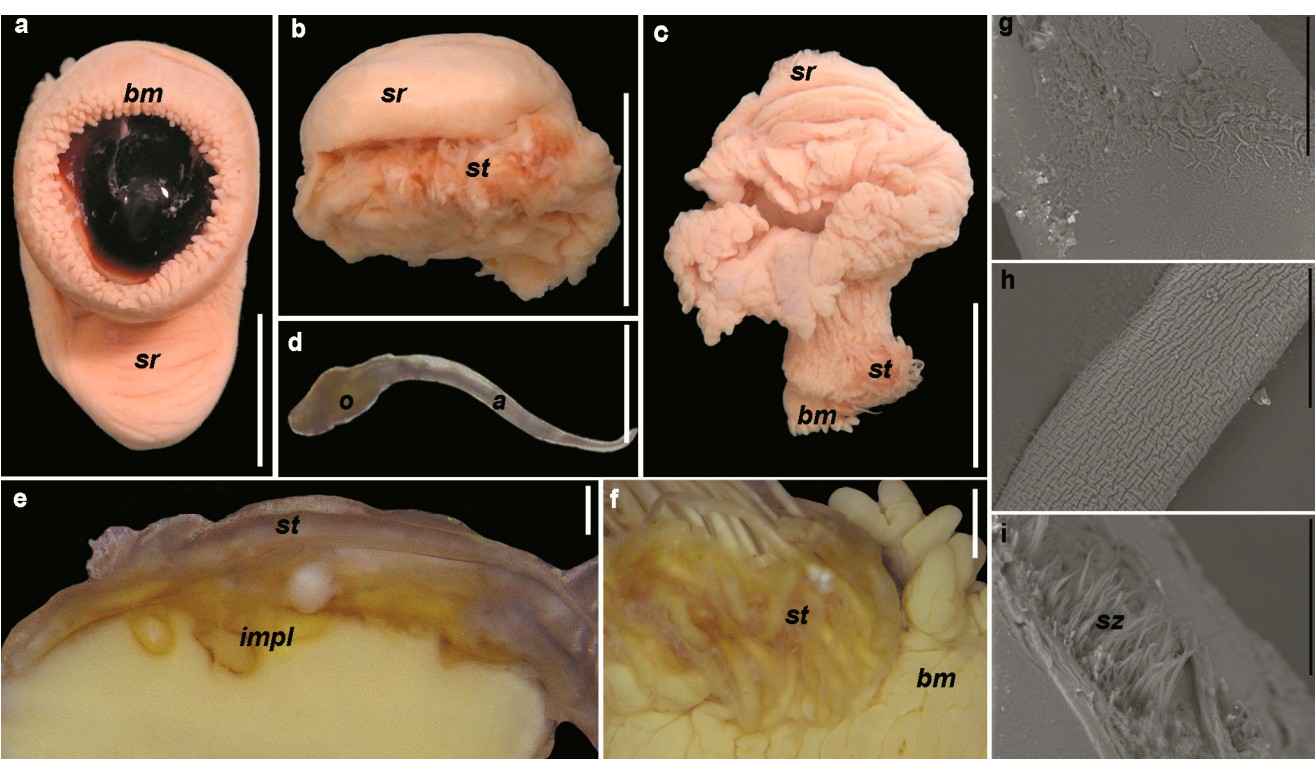

**Figure 7.** Seminal receptacle and spermatangia of *Sepia esculenta*: (**a**) Location of seminal receptacle under the buccal membrane; (**b**) Seminal receptacle external view; (**c**); (**d**) Morphology of spermatangia; (**f**) Implantation of spermatangia into the buccal membrane; (**e**) Implantation of spermatangia into the seminal receptacle; (**g**) Surface of oral part of spermatangia; (**h**) Surface of aboral part of spermatangia; (**i**) Spermatozoa in the spermatangia. *bm*: buccal membrane; *sr*: seminal receptacle; *st*: spermatangia; *o*: oral part of spermatangia; *a*: aboral part of spermatangia; *imp*: implantation of spermatangia into tissue of seminal receptacle; *sz*; spermatozoa. Scale bars: (**a–c**): 10 mm, (**d–f**): 1 mm, (**g,h**): 100 nm, (**i**): 50 μm.

## 4. Discussion

### 4.1. Female Reproductive System

The female reproductive system of *S. esculenta*, in terms of morphology, was in line with the published descriptions of the reproductive systems of other *Sepia* spp. and *Sepiella* spp. [51–57].

The PF of *S. esculenta* published prior to this study varied from 934 to 3206. Arima et al. [37] reported that the PF of the specimens collected during the spawning season was 1051–2427 (mean 1778). Yasuda [35] reported that the PF of the specimens collected during the spawning season was 2280–3206 (mean 2650). Tomiyama [36] reported that the PF the of specimens collected before the spawning season was 2280–3206 (mean 2650) and that the PF of the specimens collected during the spawning season was 934–2205 (mean 1590) [32]. Zhang et al. [38] reported that PF was 2940.6 ± 648.9 in the "early" period, 1802.1 ± 386.9 in the "middle" period, and 1593.6 ± 476.0 in the "later" period: the number of oocytes contained in the ovary was lower in the specimens collected in the mid-spawning period and later. In our study, the PF of *S. esculenta* was 1701–3719 (2945.1 ± 317.30) oocytes. Therefore, the values of the PF obtained in our study were more consistent with the values obtained for individuals collected before the spawning period in previous studies, and were the maximum PF for the species.

The decrease in the PF could be a result of oocyte resorption. The proportion of oocytes undergoing resorption in the ovary ranged from 0.6 to 13.2 (8.7 ± 1.76)% PF. A decrease in the PF due to oocyte resorption in cuttlefish had only been found in the common cuttlefish, *S. officinalis* Linnaeus, 1758 [55], to date. The PF of the maturing and pre-spawning

mature females (mean 5871) was lower than the PF of the mature spawning females (mean 3265) [26]. There were three types of oocyte resorption in cephalopods, as reviewed by Nigmatullin [58]: (1) the total type, where the whole stock of the oocyte was resorbed in the pre-spawning females because of environmental stress or developmental anomalies; (2) the final type, where the resorption of all oocytes occurred only in spent females; and (3) the regulation type, where the resorption of the protoplasmic and vitelline oocytes was a normal part of oogenesis. Regulation resorption occurred in species producing intermediate and large-sized ripe eggs (>2.5–3 mm) [58]. The oocyte resorption observed in the cuttlefish *S. officinalis* and *S. esculenta* belonged to the regulation type (Laptikhovsky et al. [26]; this study). The oocyte resorption was previously found in all phases of oocyte development. The oocyte resorption was most frequent in late previtellogenic and vitellogenic oocytes [57]. The diameters of the oocytes undergoing resorption were 0.2–5.7 (2.9 ± 0.03) mm. Most of the resorbing oocytes had a diameter of 2–4 mm (Figure 3). The mean diameters of the vitellogenic oocytes ranged from 2.5 mm at early vitellogenesis to 4.8 mm at late vitellogenesis. It could be concluded that the oocyte resorption in *S. esculenta* proceeded most intensively during the period of vitellogenesis. In contrast, the ratio of the previtellogenic oocytes undergoing resorption (<1 mm) was much lower.

The analysis of the oocyte phase distribution showed the simultaneous presence of five oocyte phases (previtellogenic, early vitellogenic, mid-vitellogenic, late vitellogenic, and ripe) within the ovary with a predominance of previtellogenic oocytes at 54.6–88.1%. The other oocyte phases occurred in the ovary in much lower quantities. The oocyte phase distribution demonstrated that the ovulation pattern of *S. esculenta* was group-synchronous. Group-synchronous ovulation has been characterised by the presence of several oocyte populations of different phases, with a predominance of one oocyte phase [58–60]. The oogenesis of *S. esculenta* was previously, and incorrectly, described as asynchronous [29]. The ovulation pattern of some other Sepiida species has also often been described as asynchronous [13,16,28]; however, observations of the oocyte phase distribution have shown the predominance of small yolkless oocytes in many *Sepia spp.*, such as *S. officinalis* [28], *S. opipara* (Iredale, 1926 [61]), *S. rozella* (Iredale, 1926 [61]), *S. plangon* (Gray, 1849 [62]) [27], *S. elegans* (Blainville, 1827 [63]) [16], and *S. orbignyana* (Ferussac [in d'Orbigny], 1826 [64]) [13]. This indicated that the ovulation patterns of the following species could also be described as group-synchronous. The summarized data of the female reproductive biology are presented in Table S6.

In cephalopods, 8 [65,66] or 9 [29,46] phases of oocyte development are distinguished, and oogenesis has been divided into 4 periods [64,65]. The gonads of mature *S. esculenta* females consisted of only seven oocyte phases (simple follicle, early yolkless, late yolkless, early vitellogenic, mid-vitellogenic, late vitellogenic, and ripe oocytes), belonging to three periods of oogenesis (protoplasmic, interstitial, and trophoplasmic growth periods). The pre-meiotic oocyte phase (the period of oogonia production) and the primary growth phase (the protoplasmic growth period) were not observed in the gonads of mature *S. esculenta* females, suggesting that they had already disappeared in earlier maturity stages. In a previous study [29], the oogenesis of *S. esculenta* was divided into nine oocyte phases. The double-follicular-cell phase, the follicle-penetration phase, the previtellogenic phase, and the mature phase were observed by Yin et al. [29] and corresponded to the simple follicle, early yolkless, late yolkless, and ripe oocyte phases, respectively, in our study. However, in a previous study, all vitellogenic oocytes filled by yolk had been classified as in the late-vitellogenic phase [29]. In our study, the vitellogenic oocytes were studied in more detail. We divided the vitellogenic oocytes into three phases (early, middle, and late vitellogenesis) based on the amount of yolk they contained.

The female reproductive system of *S. esculenta* was studied in detail for the first time. The main features of the reproductive biology of the females were as follows: (1) a PF of 1701–3719 (2945.1 ± 317.30) oocytes; (2) regulated oocyte resorption (0.6–13.2 (8.7 ± 1.76)% PF) of previtellogenic and vitellogenic oocytes; (3) a group-synchronous ovulation pattern

with a predominance of previtellogenic oocytes; and (4) an absence of the pre-meiotic and primary growth oocyte phases in the ovary of mature females.

*4.2. Male Reproductive System*

The male reproductive system of *S. esculenta*, in terms of morphology, was largely in line with the descriptions of the reproductive systems of other *Sepia* spp. And *Sepiella* spp. [51–57]. The testis and all six parts of the SG showed a tendency towards negative allometric growth, and the spermatophoric sac showed a tendency towards positive allometric growth.

The number of the spermatophores of *S. esculenta* was 146–1698 (422.7 ± 79.92). The following significant correlations were found: a positive correlation between ML and the number of spermatophores and a positive correlation between the number of the spermatophores and the total mass of the reproductive system. Therefore, larger males with a reproductive system of a higher mass usually had more spermatophores than their smaller counterparts. Two males had an extremely high number of spermatophores (1343 and 1698). It could be concluded that these males were larger than the others (the MLs and masses of their reproductive systems were not available). The number of spermatophores, excluding these males, was 146–486 (311.8 ± 24.68), which was slightly higher than the previously reported number of spermatophores at 200–300 [34].

This study provided the first data on the length and morphology of spermatophores in *S. esculenta*. The golden cuttlefish had longer spermatophores than the other Sepiida species described in previous studies (Table S5). The relative SLs had a well-expressed negative correlation with the MLs, as well as the testis and all six parts of the SGs also had a negative correlation with MLs. The spermatophore morphology is an important taxonomic characteristic in cephalopods; the ejaculatory apparatus, the cement body, and the seminal reservoir structure, as well as the morphology of the connection between the cement body and the seminal reservoir are the most important taxonomic characteristics in spermatophore morphology [67]. The main differences in the spermatophore structures between the *S. esculenta* and the other *Sepia* spp. and *Sepiella* spp. are the morphologies of their cement bodies. The cement body of the *S. esculenta* consisted of conical oral and cylindrical aboral parts, and the aboral part was approximately 1.5 times longer than the oral one. This resembled the cement body structures in the following species: *S. shazae* (Lipiński and Leslie 2018 [68]), *S. barosei* (Lipiński 2020 [69]), *S. roeleveldi* (Lipiński 2020 [69]), *S. pulchra* (Roeleveld and Liltved 1985 [70]), *S. plana* (Lu and Reid 1997 [51]), *S. senta* (Lu and Reid 1997 [51]), *S. grahami* (Reid 2001 [54]), *S. plathyconchalis* (Fillipova and Khromov, 1991 [71]), *S. typica* (Steenstrup, 1875 [72]) [73], *Sepiella mangkangunga* (Reid and Lu, 1998 [56]), and *Sep. weberi* (Adam 1939 [74]) [56]. The morphology of the cement body in *S. stellifera* (Homenko and Khromov 1984 [75]) resembled this structure to a lesser degree, as its aboral part was much shorter and flask shaped. The morphology of the cement body in other species clearly differed from this structure. The cement bodies of *S. ivanovi* (Khromov 1982 [76]), *S. mascarensis* (Fillipova and Khromov 1991 [71]), *S. sokotriensis* (Khromov 1988 [77]), and *S. mirabilis* (Khromov 1988 [77]) consisted of 3 parts, and the morphologies of these parts differed sharply among these species. In contrast, the cement bodies of *S. hedleyi* Berry 1918 [78]) [55] and *S. mestus* (Gray 1849 [62]) [57] were not bipartite. Therefore, three types of structures of the cement body were found in different *Sepia* spp. and *Sepiella* spp.

There is a tendency in coleoid cephalopods to show an ontogenetic increase in the length of the spermatophore: The spermatophores produced earlier in ontogenesis are longer than those produced later in ontogenesis. The oldest and smallest spermatophores are usually contained in the penis, while the youngest and largest spermatophores are contained in the fundus of the spermatophoric sac. This tendency has been repeatedly observed in squids and other sepiolid in previous studies [46,47,79–86] and was recently found in Cirrata [87]. This study was the first to provide data on the ontogenetic changes of the spermatophore size in cuttlefish. The males collected at the beginning of the spawning season clearly demonstrated the ontogenetic increase in spermatophore length from the

penis to the fundus. Additionally, the spermatophore mass; the seminal reservoir length, width, and volume; and the cement body length increased during ontogenesis. Therefore, the quality of the spermatophore improved due to an increase in the sperm volume in the seminal reservoir and its adhesive ability. In contrast, the males collected at the end of the spawning season had younger spermatophores from the fundus that were shorter than older spermatophores from the central portion of the spermatophoric sac. Moreover, the length, width, and volume of the seminal reservoir and the spermatophore mass were smaller in the fundus than in the central part. At the same time, the length of the head, the ejaculatory apparatus, and the cement body decreased during ontogenesis. The formation of smaller spermatophores with the reduced size of the sperm reservoir is characteristic of residual spermatophorogenesis. Residual spermatophorogenesis is also characterised by the degeneration of the testis and decreasing function of the reproductive system. The males at the stage of residual spermatophorogenesis still had enough spermatophores in the spermatophoric sac for copulation, but the spermatophore morphology resembled that of false spermatophores [82,88]. The studied males collected at the end of the spawning season were characterised by the formation of smaller spermatophores with a reduced size in their sperm reservoir but did not have other signs of reproductive system degeneration. Their spermatophore morphology was typical of true spermatophorogenesis. Consequently, there was a gradual transition from a true spermatophorogenesis to a residual form in males at the end of the spawning season. A similar process of decreasing sperm mass in the seminal reservoir at the final phase of functional spermatophorogenesis was also described in squids [83].

The male reproductive system of *S. esculenta* was studied in detail for the first time. The main features of the reproductive biology of males were as follows: (1) the negative allometric growth of the testis and all six parts of the SG; (2) the number of spermatophores at 146–1698 (422.7 ± 79.92); (3) an increase in the number of the spermatophores with an increasing ML and mass of the reproductive system; (4) the longest spermatophores among the studied Sepiida species; (5) the spermatophore's characteristic cement body consisting of short oral and long aboral parts; (6) the ontogenetic increase in the length of the spermatophores and the spermatophore mass, as well as the seminal reservoir length, width, and volume at the beginning of the spawning season as typical feature of true spermatophorogenesis; (7) the improvement of the quality of the spermatophore due to an increase in sperm volume in the seminal reservoir and its adhesive ability during ontogenesis; and (8) the ontogenetic decrease of the SL; the width, the length, and the volume of the seminal reservoir; and spermatophore mass during the end of the spawning season as a feature of gradual transition to residual spermatophorogenesis.

The spermatangia of *S. esculenta* had relatively well-defined oral and aboral parts. The spermatangia were implanted on the area of the seminal receptacles, as in other studies on *Sepia* spp. [41,89,90]. Additionally, the location of the spermatangia on the buccal membrane was observed. This location of the spermatangia was previously described in S. apama (Gray, 1849 [62]) [89]. The implantation of the spermatangia was shallow, i.e., only the base of the spermatangia had penetrated the female tissues. This implantation type was found in other *Sepia* spp., as well as in other groups of Decapodiformes [91].

## 5. Conclusions

In summary, we studied the reproductive biology of *S. esculenta* in detail, and for the first time for this species, the following data were obtained: (1) the morphology of male and female reproductive systems; (2) oocyte resorption; (3) the absence of the pre-meiotic and the primary growth oocyte phases in the ovary of mature females; (4) the allometric growth of the parts of the male reproductive system; (5) the detailed spermatophore measurements and the morphology; and (6) the spermatangia number, length, and implantation pattern. The ontogenetic changes in the length of the spermatophore and its parts were described for the first time in the order Sepiida. The ontogenetic improvement of the quality of the spermatophore due to an increase in the sperm volume in the seminal reservoir and its

adhesive ability were also described for the first time in the order Sepiida. The data obtained in the present study significantly increase and correct the existing knowledge on *S. esculenta* biology. Moreover, these data help to explain the general patterns of reproductive biology in cuttlefish, and in Cephalopoda as a whole.

**Supplementary Materials:** The following supporting information can be downloaded at: https://www.mdpi.com/article/10.3390/d15030455/s1, Table S1. Studied specimens of *Sepia esculenta*. ML–mantle length. Table S2. Measurements of the female and male reproductive system parts in *Sepia esculenta*. ML–mantle length; SG–spermatophoric gland. The values are min–max (mean ± SE). Table S3. Equations of correlations between mantle length and different parts of male reproductive system in *Sepia esculenta*. ML–mantle length; SG–spermatophoric gland. Significant *p*-values are in bold. Table S4. Equations of correlations between mantle length and spermatophore length, and correlations between spermatophore length and different parts of spermatophore. ML–mantle length, SL–spermatophore length. Significant *p*-values are in bold [92]. Table S5. Spermatophore length and number of spermatophores of the different *Sepia* spp. And *Sepiella* spp [93]. Table S6. Some features of cuttlefish females reproductive biology. Figure S1. Correlations between mantle length and different parts of male reproductive system in *Sepia esculenta*. ML–mantle length; SG–spermatophoric gland. Figure S2. Correlation between number of spermatophores and ML. ML–mantle length. Figure S3. Correlation between number of spermatophores and weight of reproductive system. Figure S4. Correlation between number of spermatophores and total volume of sperm. Figure S5. Correlations between absolute/relative spermatophore length and ML. ML–mantle length, SL–spermatophore length. Figure S6. Correlations between spermatophore length and different parts of spermatophore. ML–mantle length, SL–spermatophore length.

**Author Contributions:** Conceptualization, E.V.V., A.V.G. and R.M.S.; methodology, E.V.V., A.V.G. and R.M.S.; formal analysis, E.V.V. and A.V.G.; writing—original draft preparation, E.V.V.; writing—review and editing, E.V.V., A.V.G. and R.M.S. All authors have read and agreed to the published version of the manuscript.

**Funding:** The authors acknowledge the financial support received from the Okayama University, which allowed to collect samples.

**Institutional Review Board Statement:** All applicable international, national, and/or institutional guidelines for animal testing, animal care and use of animals were followed by the authors.

**Data Availability Statement:** All relevant data are included in the paper and/or in the Supplementary Information.

**Acknowledgments:** We are grateful to Masayuki Saigusa for inviting Rushan M. Sabirov and Alexey V. Golikov to Japan, where they had an opportunity to collect the samples; to Oleg A. Gusev for providing part of the samples; to Andrey G. Porfiriev and Dayana N. Sharafutdinova for help with the histology; to Azat G. Kadirov and Islam R. Nigmetzyanov for help with the microscopy; and to Alexander A. Novikov for their advice on vector graphics.

**Conflicts of Interest:** The authors declare that they have no conflict of interest.

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
