# Peer review of "Reproductive Biology of the Golden Cuttlefish Sepia esculenta (Cephalopoda, Sepiida)"

_diversity, doi:10.3390/d15030455_

Round 1

Reviewer 1 Report

I enjoyed reading the manuscript! I only have one remark regarding Figure 4 - the staining of the section seems to be a bit too red and also it looks as if there might have been a slight astigmatism on the image. this might be either due to the thickness of the section or the setting of the microscope. 

My only suggestion to improve this paper would be to imporve this figure.

thank you very much for this publication on this important species!

Author Response

We thank the Reviewer for thoroughly reading our manuscript, and provide our replies below.

Point 1: I only have one remark regarding Figure 4 - the staining of the section seems to be a bit too red and also it looks as if there might have been a slight astigmatism on the image. this might be either due to the thickness of the section or the setting of the microscope.

Response 1: Figures 1a, 1b and 1c were replaced with a better images. Unfortunately, the red color is the result of staining and cannot be changed.

Reviewer 2 Report

Reproductive biology of the golden cuttlefish Sepia esculenta (Cephalopoda, Sepiida)

Review

From my perspective, making this manuscript a fair review is hampered by some major flaws of its own.

The manuscript would put forward arguments to be considered as patterns to explain the reproductive biology in the species and the taxon as a whole.

To go through the outputs from the range and the specifics of the sampling may require questioning refinement and ensure consistency in the procedures applied.

Sampling statement : N= 6 females, retrieved from different locations, same for males except for n=25, those latest exhibiting larger body size variance. What indication for uneven sex ratio in fishing catches depending on the sites. Were all the individuals collected at the same timescale in the spawning season (see l.505) ?

Despite the maturity scale states fine description, the within sample distribution was not provided. It is advises to either use or remove the long description list.

It appears that diversity and range would be a key factor to drive one component of the analysis. Reading the figures that support inter and intra-individual diversity along the text is challenging (Aren't they in the tables). Summarized histograms might be a fruitful alternative to support results description? A synthesis chart might also help providing the species position in the group regarding the reproductive biology. In any case, the process leading up to the calculation of the mean and its deviation needs to be clarified.

Figures : A major concern regarding a biological report, the quality of the pictures should be improved, avoiding post-processing and with limited captions superimposed sometimes ( e.g. fig.5)

The literature reference could be synthesized as well. There are 34 references before introducing the species of interest, subject of the work.

Round 2

Reviewer 2 Report

No further comment from me